# EFFECTIVE OFFLINE REINFORCEMENT LEARNING VIA CONSERVATIVE STATE VALUE ESTIMATION

## ABSTRACT

Offline RL seeks to learn effective policies solely from historical data, which expects to perform well in the online environment. However, it faces a major challenge of value over-estimation introduced by the distributional drift between the dataset and the current learned policy, leading to learning failure in practice. The common approach is adding a penalty term to reward or value estimation in the Bellman iterations, which has given rise to a number of successful algorithms such as CQL. Meanwhile, to avoid extrapolation on unseen states and actions, existing methods focus on conservative Q-function estimation. In this paper, we propose CSVE, a new approach that learns conservative V-function via directly imposing penalty on out-of-distribution states. We prove that for the evaluated policy, our conservative state value estimation satisfies: (1) over the state distribution that samples penalizing states, it lower bounds the true values in expectation, and (2) over the marginal state distribution of data, it is no more than the true values in expectation plus a constant decided by sampling error. Further, we develop a practical actor-critic algorithm in which the critic does the conservative value estimation by additionally sampling and penalizing the states *around* the dataset, and the actor applies advantage weighted updates to improve the policy. We evaluate in classic continual control tasks of D4RL, showing that our method performs better than the conservative Q-function learning methods (e.g., CQL) and is strongly competitive among recent SOTA methods.

# 1 INTRODUCTION

Reinforcement Learning (RL), which learns to act by interacting with the environment, has achieved remarkable success in various tasks. However, in most real applications, it is impossible to learn online from scratch as exploration is often risky and unsafe. Instead, offline RL((Fujimoto et al., 2019; Lange et al., 2012)) avoids this problem by learning the policy solely from historical data. However, the naive approach, which directly uses online RL algorithms to learn from a static dataset, suffers from the problems of value over-estimation and policy extrapolation on OOD (out-of-distribution) states or actions.

Recently, conservative value estimation, being conservative on states and actions where there are no enough samples, has been put forward as a principle to effectively solve offline RL ((Shi et al., 2022; Kumar et al., 2020; Buckman et al., 2020)). Prior methods, e.g., Conservative Q-Learning (CQL Kumar et al. (2020)), avoid the value over-estimation problem by systematically underestimating the Q values of OOD actions on the states in the dataset. In practice, it is often too pessimistic and thus leads to overly conservative algorithms. COMBO (Yu et al., 2021) leverages a learnt dynamic model to augment data in an interpolation way, and then learn a Q function that is less conservative than CQL and derives a better policy in potential.

In this paper, we propose CSVE(Conservative State Value Estimation), a new offline RL approach. Unlike the above traditional methods that estimate conservative values by penalizing Q-function on OOD states or actions, CSVE directly penalizing the V-function on OOD states. We prove in theory that CSVE has tighter bounds on true state values than CQL, and same bounds as COMBO but under more general discounted state distributions which leads to more space for algorithm design. Our main contributions are as follows.

- The conservative state value estimation with related theoretical analysis. We prove that it lower bounds the real state values in expectation over any state distribution that is used to sample OOD states, and is up-bounded by the real values in expectation over the marginal state distribution of the dataset plus a constant term depending on only sampling errors. Compared to prior work, it has several advantages to derive a better policy in potential.

- A practical Actor-Critic implementation. It approximately estimates the conservative state values in the offline context and improves the policy via advantage weighting updates. In particular, we use a dynamics model to generalize over in-distribution space and sample OOD states that are directly reachable from the dataset.

- Experimental evaluation on continuous control tasks of Gym (Brockman et al., 2016) and Adroit (Rajeswaran et al., 2017) in D4RL (Fu et al., 2020) benchmarks, showing that CSVE performs better than prior methods based on conservative Q-value estimation, and is strongly competitive among main SOTA offline RL algorithms.

## 2 PRELIMINARIES

**Offline Reinforcement Learning.** Consider the Markov Decision Process $M := (\mathcal{S}, \mathcal{A}, P, r, \rho, \gamma)$, which consists of the state space $\mathcal{S}$, the action space $\mathcal{A}$, the transition model $P : \mathcal{S} \times \mathcal{A} \to \Delta(\mathcal{S})$, the reward function $r : \mathcal{S} \times \mathcal{A} \to \mathbb{R}$, the initial state distribution $\rho$ and the discount factor $\gamma \in (0, 1]$. A stochastic policy $\pi : \mathcal{S} \to \Delta(\mathcal{A})$ takes an action in probability given the current state. A transition is the tuple $(s_t, a_t, r_t, s_{t+1})$ where $a_t \sim \pi(\cdot|s_t)$, $s_{t+1} \sim P(\cdot|s_t, a_t)$ and $r_t = r(s_t, a_t)$. We assume that the reward values satisfy $|r(s, a)| \leq R_{max}, \forall s, a$. A trajectory under $\pi$ is the random sequence $\tau = (s_0, a_0, r_0, s_1, a_1, r_1, \ldots, s_T)$ which consists of continuous transitions starting from $s_0 \sim \rho$.

The standard RL is to learn a policy $\pi \in \Pi$ that maximize the future cumulative rewards $J_\pi(M) = \mathbb{E}_{M,\pi}[\sum_{t=0}^\infty \gamma^t r_t]$ via active interaction with the environment $M$. At any time $t$, for the policy $\pi$, the value function of state is defined as $V^\pi(s) := \mathbb{E}_{M,\pi}[\sum_{k=0}^\infty \gamma^{t+k} r_{t+k}|s_t = s]$, and the Q value function is $Q^\pi(s, a) := \mathbb{E}_{M,\pi}[\sum_{k=0}^\infty \gamma^{t+k} r_{t+k}|s_t = s, a_t = a]$. The Bellman operator is a function projection: $\mathcal{B}^\pi Q(s, a) := r(s, a) + \gamma \mathbb{E}_{s' \sim P(\cdot|s,a), a' \sim \pi(\cdot|s')}[Q(s', a')]$, or $\mathcal{B}^\pi V(s) := \mathbb{E}_{a \sim \pi(\cdot|s)}[r(s, a) + \gamma \mathbb{E}_{s' \sim P(\cdot|s,a)}[V(s')]]$, which leads to iterative value updates. Bellman consistency implies that $V^\pi(s) = \mathcal{B}^\pi V^\pi(s), \forall s$ and $Q^\pi(s) = \mathcal{B}^\pi Q^\pi(s, a), \forall s, a$. In practice with function approximation, we use the empirical Bellman operator $\hat{\mathcal{B}}^\pi$ where the former expectations are estimated with data.

The offline RL is to learn the policy $\pi$ from a static dataset $D = \{(s, a, r, s')\}$ consisting of transitions collected by any behaviour policy, aiming to behave well in the online environment. Note that, unlike the standard online RL, offline RL cannot interact with the environment during learning.

**Conservative Value Estimation.** One main challenge in offline RL is the over-estimation of values introduced by extrapolation on unseen states and actions, which may make the learned policy collapse. To address this issue, conservatism or pessimism are used in value estimation, e.g. CQL learns a conservative Q-value function by penalizing the value of unseen actions on states:

$$\hat{Q}^{k+1} \leftarrow \arg\min_Q \; \alpha \left( \mathbb{E}_{s \sim D, a \sim \mu(a|s)}[Q(s, a)] - \mathbb{E}_{s \sim D, a \sim \hat{\pi}_\beta(a|s)}[Q(s, a)] \right)$$
$$+ \frac{1}{2} \mathbb{E}_{s,a,s' \sim D}[(Q(s, a) - \hat{\mathcal{B}}_\pi \hat{Q}^k(s, a))^2] \tag{1}$$

where $\hat{\pi}_\beta$ and $\pi$ are the behaviour policy and learnt policy separately, $\mu$ is any arbitrary policy different from $\hat{\pi}_\beta$, and $\alpha$ the factor for trade-off of conservatism.

**Constrained Policy Optimization.** To address the issues of distribution drift between learning policy and behaviour policy, one approach is to constrain the learning policy close to the behaviour policy (Bai et al., 2021; Wu et al., 2019; Nair et al., 2020; Levine et al., 2020; Fujimoto et al., 2019). Here we take Advantage Weighted Regression(Peng et al. (2019b); Nair et al. (2020)) which adopts an implicit KL divergence to constrain the distance of policies as example:

$$\pi^{k+1} \leftarrow \arg\max_\pi \mathbb{E}_{s,a \sim D}\left[\log \pi(a|s) \frac{1}{Z(s)} \exp\left(\frac{1}{\lambda} A^{\pi^k}(s, a)\right)\right] \tag{2}$$

where $A^{\pi^k}$ is the advantage of policy $\pi^k$, and $Z$ the normalization constant for $s$.

**Model-based Offline RL.** In RL, the model is an approximation of the MDP $M$. We denote a model as $\hat{M} := (\mathcal{S}, \mathcal{A}, \hat{P}, \hat{r}, \rho, \gamma)$, where $\hat{P}$ and $\hat{r}$ are approximations of $P$ and $r$ respectively. In the setting of offline RL, the model is used to roll out and augment data (Yu et al., 2020; 2021) or act as a surrogate of real environment to interact with agent (Kidambi et al., 2020). In this paper, we use model to sample the next states that are approximately reachable from the dataset.

## 3 CONSERVATIVE STATE VALUE ESTIMATION

In the offline setting, the value overestimation is a major problem resulting in failure of learning a reasonable policy (Levine et al., 2020; Fujimoto et al., 2019). In contrast to prior works(Kumar et al., 2020; Yu et al., 2021) that get conservative value estimation via penalizing Q function for OOD state-action pairs , we directly penalize V function for OOD states. Our approach provides several novel theoretic results that allow better trade-off of conservative value estimation and policy improvement. All proofs of our theorems can be found in Appendix A.

### 3.1 CONSERVATIVE OFF-POLICY EVALUATION

Our approach is an alternative approach to CQL(Kumar et al., 2020). Instead of learning a conservative Q function, we aim to conservatively estimate the value $V^\pi(s)$ of a target policy $\pi$ given a dataset $D$ to avoid overestimation of out-of-distribution states. To achieve this, we penalize the V-values evaluated on states that is more likely to be out-of-distribution and pushing up the V-values on states that is in the distribution of the dataset, which is achieved through the following iteration:

$$\hat{V}^{k+1} \leftarrow \arg\min_V \frac{1}{2}\mathbb{E}_{s\sim d_u(s)}[(\hat{\mathcal{B}}^\pi \hat{V}^k(s) - V(s))^2] + \alpha(\mathbb{E}_{s'\sim d(s)}V(s') - \mathbb{E}_{s\sim d_u(s)}V(s)) \quad (3)$$

where $d_u(s)$ is the discounted state distribution of D, $d(s)$ is any state distribution, and $\hat{\mathcal{B}}^\pi$ is the empirical Bellman operator (see appendix for more details). Considering the setting without function approximation, by setting the derivative of Eq. 3 as zero, the V function found by approximate dynamic programming in iteration $k$ can be obtained:

$$\hat{V}^{k+1}(s) = \hat{\mathcal{B}}^\pi \hat{V}^k(s) - \alpha[\frac{d_{(s)}}{d_u(s)} - 1], \quad \forall s, k. \quad (4)$$

Denote the function projection on $\hat{V}^k$ in Eq. 4 as $\mathcal{T}^\pi$. We have Lemma 1, and thus $\hat{V}^k$ converges to a unique fixed point.

**Lemma 1.** *For any $d$ with $\operatorname{supp} d \subseteq \operatorname{supp} d_u$, $\mathcal{T}^\pi$ is a $\gamma$-contraction in $L_\infty$ norm.*

**Theorem 1.** *For any $d$ with $\operatorname{supp} d \subseteq \operatorname{supp} d_u$ ($d \neq d_u$), with a sufficiently large $\alpha$ (i.e., $\alpha \geq \mathbb{E}_{s\sim d(s)}\mathbb{E}_{a\sim\pi(a|s)}\frac{C_{r,t,\delta}R_{max}}{(1-\gamma)\sqrt{|D(s,a)|}}/\mathbb{E}_{s\sim d(s)}[\frac{d(s)}{d_u(s)} - 1])$), the expected value of the estimation $\hat{V}^\pi(s)$ under $d(s)$ is the lower bound of the true value, that is: $\mathbb{E}_{s\sim d(s)}[\hat{V}^\pi(s)] \leq \mathbb{E}_{s\sim d(s)}[V^\pi(s)]$.*

$\hat{V}^\pi(s) = \lim_{k\to\infty} \hat{V}^k(s)$ is the converged value estimation with the dataset $D$, and $\frac{C_{r,t,\delta}R_{max}}{(1-\gamma)\sqrt{|D(s,a)|}}$ is related to sampling error introduced by the use empirical rather than Bellman operator. If the counts of each state-action pair is greater than zero, $|D(s,a)|$ denotes a vector of size $|\mathcal{S}||\mathcal{A}|$ containing counts for each state-action pair. If the counts of this state action pair is zero, the corresponding $\frac{1}{\sqrt{|D(s,a)|}}$ is large but finite value. We assume that with probability $\geq 1 - \delta$, the sampling error is less than $\frac{C_{r,t,\delta}R_{max}}{(1-\gamma)\sqrt{|D(s,a)|}}$, while $C_{r,t,\delta}$ is a constant (See appendix for more details.) Note that if the sampling error is ignorable, $\alpha > 0$ can guarantee the lower bound results.

**Theorem 2.** *The expected value of the estimation $\hat{V}^\pi(s)$ under the state distribution of the original dataset is the lower bound of the true value plus the term of irreducible sampling error, that is: $\mathbb{E}_{s\sim d_u(s)}[\hat{V}^\pi(s)] \leq \mathbb{E}_{s\sim d_u(s)}[V^\pi(s)] + \mathbb{E}_{s\sim d_u(s)}(I - \gamma P^\pi)^{-1}\mathbb{E}_{a\sim\pi(a|s)}\frac{C_{r,t,\delta}R_{max}}{(1-\gamma)\sqrt{|D(s,a)|}}.$*

, where $P^\pi$ refers to the transition matrix coupled with policy $\pi$ (see Appendix for details).

Now we show that, during iterations, the gap between the value of in-distribution state and out-of-distribution state in the estimated V-function is higher than in the true V-functions.

**Theorem 3.** *At any iteration $k$, with a large enough $\alpha$, our method expands the difference in expected V-values under the chosen state distribution and the dataset state distribution, that is:*
$$\mathbb{E}_{s\sim d_u(s)}[\hat{V}^k(s)] - \mathbb{E}_{s\sim d(s)}[\hat{V}^k(s)] > \mathbb{E}_{s\sim d_u(s)}[V^k(s)] - \mathbb{E}_{s\sim d(s)}[V^k(s)].$$

In the policy extraction part, this property enables our policy to take actions $a$ in state $s(s \sim D)$ that remains in distribution instead of out of distribution, given that our estimated V-function does not overestimate the erroneous out-of-distribution states compared to the in-distribution states.

Now we present four remarks to explain how the above theorems guide applications of Eq. 3 in offline RL algorithms.

**Remark 1.** In Eq. 3, if $d = d_u$, the penalty on out-of-distribution states degenerates, which means that the policy should not reach states with low support in data, and consequently never explore the unseen actions at the state. Indeed, AWAC Nair et al. (2020) adopts this setting. We show that with proper choice of $d$ different from $d_u$, our method performs better than AWAC in practice.

**Remark 2.** Theorem 2 implies that under $d_u$, the marginal state distribution of data, the expectation estimated value of $\pi$ should either be lower than the true value, or higher than the true value but within a threshold. This fact motivates our advantage weighted policy update method in Eq. 11.

**Remark 3.** Theorem 1implies that under $d$, say the discounted state distribution of any policy, the expectation estimated value of $\pi$ should lower bounds the true value. This fact motivates our policy improvement method of unifying advantage weighted update with a bonus exploration in Eq. 12.

**Remark 4.** Theorem 3 states $\mathbb{E}_{s\sim d(s)}[V^k(s)] - \mathbb{E}_{s\sim d(s)}[\hat{V}^k(s)] > \mathbb{E}_{s\sim d_u(s)}[V^k(s)] - \mathbb{E}_{s\sim d_u(s)}[\hat{V}^k(s)]$. That is to say, under the distribution $d$, the amount of value under-estimation in expectation is larger than that of the behaviour policy $d_u$. With proper choice of $d$, it is safe and effective to derive a new and potentially better policy with $\hat{V}^k$. Our algorithm choose the distribution of model predictive next-states as $d$, i.e., $s' \sim d$ is implemented by $s \sim D, a \sim \pi(\cdot|s), s' \sim \hat{P}(\cdot|s,a)$, which effectively builds a soft 'river' with low values around the dataset.

**Comparison with prior work:** CQL (Eq.1), which penalizes Q-function of OOD actions on states in history data, guarantees the lower bounds on state-wise value estimation: $\hat{V}^\pi(s) = E_{\pi(a|s)}(\hat{Q}^\pi(s,a)) \leq E_{\pi(a|s)}(Q^\pi(s,a)) = V^\pi(s)$ for all $s \in D$. COMBO, which penalizes Q-function of OOD states and actions of an interpolation of history data and model-based roll-outs, guarantees the lower bound of state value expectation: $\mathbb{E}_{s\sim\mu_0}[\hat{V}^\pi(s)] \leq \mathbb{E}_{s\sim\mu_0}[V^\pi(s)]$ where $\mu_0$ is the initial state distribution (Remark 1, section A.2 of COMBO Yu et al. (2021)); which is a special case of our result in Theorem 1 when $d = \mu_0$. Although both CSVE and COMBO intend to get better performance by releasing conservative estimation guarantee from the state-wise values to expectation of state values, CSVE get the same lower bounds but under more general state distribution. This provide more flexible space for algorithm design, and it is also one main reason of penalizing on $V$ rather than Q. By controlling distance of $d$ to the behaviour policy's discounted state distribution $d_\beta$, CSVE has the potential of more performance improvement.

Note that bounding $E[V[s]]$, rather than state-wise $V(s)$, would introduce a more adventurous policy, which would achieves better performance in in-distribution states and have more risk behaviors in OOD states. To deal with that limitation, we introduce a deep ensemble dynamic model to sample the OOD states for better estimation.

## 3.2 SAFE POLICY IMPROVEMENT GUARANTEES

Following prior works (Laroche et al. (2019); Kumar et al. (2020); Yu et al. (2021)), we show that our method has the safe policy improvement guarantees against the data-implied behaviour policy. We first show that our method optimizes a penalized RL empirical objective:

**Theorem 4.** *Let $\hat{V}^\pi$ be the fixed point of Equation 3, then $\pi^*(a|s) = \arg\max_\pi \hat{V}^\pi(s)$ is equivalently obtained by solving:*

$$\pi^*(a|s) \leftarrow \arg\max_\pi J(\pi, \hat{M}) - \alpha\frac{1}{1-\gamma}\mathbb{E}_{s\sim d_{\hat{M}}^\pi(s)}[\frac{d(s)}{d_u(s)} - 1] \tag{5}$$

Building upon Theorem 4, we show that our method provides a $\zeta$-safe policy improvement over $\pi_\beta$

**Theorem 5.** *Let $\pi^*(a|s)$ be the policy obtained in Equation 5. Then, it is a $\zeta$-safe policy improvement over $\hat{\pi}^\beta$ in the actual MDP M, i.e., $J(\pi^*, M) \geq J(\hat{\pi}^\beta, M) - \zeta$ with high probability 1- $\delta$, where $\zeta$ is given by:*

$$\zeta = 2(\frac{C_{r,\delta}}{1-\gamma} + \frac{\gamma R_{max}C_{T,\delta}}{(1-\gamma)^2})\mathbb{E}_{s\sim d_{\hat{M}}^\pi(s)}[\frac{\sqrt{|\mathcal{A}|}}{\sqrt{|\mathcal{D}(s)|}}\sqrt{\mathbb{E}_{a\sim\pi(a|s)}(\frac{\pi(a|s)}{\pi_\beta(a|s)})}] - \underbrace{(J(\pi^*, \hat{M}) - J(\hat{\pi}_\beta, \hat{M}))}_{\geq\alpha\frac{1}{1-\gamma}\mathbb{E}_{s\sim d_{\hat{M}}^\pi(s)}[\frac{d(s)}{d_u(s)}-1]}. \tag{6}$$

## 4 METHODOLOGY

In this section, we propose a practical Actor-Critic method for computing conservative value estimation function by approximately solving Equation 3 and taking advantage weighted policy updates. It is mainly motivated by the theoretic results, as explained by the four remarks in section 3.1. Besides, the full algorithm of deep learning implementation is presented in Appendix B.

### 4.1 CONSERVATIVE VALUE ESTIMATION

Given the access to a dataset $D$ collected by some behaviour policy $\pi_\beta$, our aim is to estimate the value function $V^\pi$ for a target policy $\pi$. As stated in section 3, to prevent the value overestimation, we instead learn a conservative value function $\hat{V}^\pi$ that lower bounds the real values of $\pi$ by adding a penalty on out-of-distribution states into the flow of Bellman projections. Our method consists of the following iterative updates of Equations 7- 9, where $\overline{\hat{Q}^k}$ is the target network of $\hat{Q}^k$.

$$\hat{V}^{k+1} \leftarrow \arg\min_V L_V^\pi(V; \overline{\hat{Q}^k}) = \alpha\left(\mathbb{E}_{s\sim D, a\sim\pi(\cdot|s), s'\sim\hat{P}(s,a)}[V(s')] - \mathbb{E}_{s\sim D}[V(s)]\right) \\ + \mathbb{E}_{s\sim D}\left[(\mathbb{E}_{a\sim\pi(\cdot|s)}[\overline{\hat{Q}^k}(s,a)] - V(s))^2\right] \tag{7}$$

$$\hat{Q}^{k+1} \leftarrow \arg\min_Q L_Q^\pi(Q; \hat{V}^{k+1}) = \mathbb{E}_{s,a,s'\sim D}\left[\left(r(s,a) + \gamma\hat{V}^{k+1}(s') - Q(s,a)\right)^2\right] \tag{8}$$

$$\overline{\hat{Q}^{k+1}} \leftarrow \omega\overline{\hat{Q}^k} + (1-\omega)\hat{Q}^{k+1} \tag{9}$$

The RHS of Eq. 7 is an approximation of Eq. 3, where the first term gives out-of-distribution states a penalty, and the second term follows the definition of V values and Q values. In Eq. 8, the RHS is TD errors estimated on transitions in the dataset $D$. Note that the target term here uses the sum of the immediate reward $r(s,a)$ and the next step state's value $\hat{V}^{k+1}(s')$. In Eq. 9, the target Q values are updated with a soft interpolation factor $\omega \in (0,1)$. $\overline{\hat{Q}^k}$ changes slower than $\hat{Q}^k$, which makes the TD error estimation in Eq. 7 more stable.

**Constrained policy.** Note that in RHS of Eq. 7, we use $a \sim \pi(\cdot|s)$ in expectation. To safely estimate the target value of $V(s)$ by $\mathbb{E}_{a\sim\pi(\cdot|s)}[\overline{\hat{Q}}(s,a)]$, almost always requires $\text{supp}(\pi(\cdot|s)) \subset \text{supp}(\pi_\beta(\cdot|s))$. We achieves this by the *advantage weighted policy update*, which forces $\pi(\cdot|s)$ have significant probability mass on actions taken by $\pi_\beta$ in data, as detailed in section 3.2.

**Model-based OOD state sampling.** In Eq. 7, we implement the state sampling process $s' \sim d$ in Eq. 3 as a flow of $\{s \sim D; a \sim \pi(a|s); s' \sim \hat{P}(s'|s,a)\}$, that is the distribution of the predictive next-states from $D$ by following $\pi$. It is beneficial in practice. On one hand, this method is efficient to sample only the states that are approximately reachable from $D$ by one step, rather than to sample

the whole state space. On the other hand, we only need the model to do one-step prediction such that no bootstrapped errors due to long horizon are introduced. Following previous work (Janner et al., 2019; Yu et al., 2020; 2021), we implement the probabilistic dynamics model using an ensemble of deep neural networks $\{p\theta^1, \ldots, p\theta^B\}$. Each neural network produces a Gaussian distribution over the next state and reward: $P_\theta^i(s_{t+1}, r | s_t, a_t) = \mathcal{N}(u_\theta^i(s_t, a_t), \sigma_\theta^i(s_t, a_t))$.

**Adaptive penalty factor $\alpha$.** The pessimism level is controlled by the parameter $\alpha \geq 0$. In practice, we set $\alpha$ adaptive during training as follows, which is similar as that in CQL(Kumar et al. (2020))

$$\max_{\alpha \geq 0} [\alpha(\mathbb{E}_{s' \sim d}[V_\psi(s')] - \mathbb{E}_{s \sim D}[V_\psi(s)] - \tau)] \tag{10}$$

, where $\tau$ is a budget parameter. If the expected difference in V-values is less than $\tau$, $\alpha$ will decrease. Otherwise, $\alpha$ will increase and penalize the out of distribution state values more aggressively.

**Discussion:** As stated in former sections, our method focuses on estimating conservative state value for learning a policy. The effectiveness of adding conservatism on $V$ function are two folds. First, penalizing $V$ values is with a smaller hypothesis space than penalizing $Q$, which would reduce the computation complexity. Second, penalizing $V$ values can achieve a more relaxed lower bound than penalizing $Q$ with ignoring the explicitly marginalization on Q values. A more relaxed lower bound guarantees more opportunities on achieving better policy.

## 4.2 ADVANTAGE WEIGHTED POLICY UPDATES

After learning the conservative $\hat{V}^{k+1}$ and $\hat{Q}^{k+1}$ (or $\hat{V}^\pi$ and $\hat{Q}^\pi$ when converged), we improve the policy by the following advantage weighted policy update (Nair et al., 2020).

$$\pi \leftarrow \arg\min_{\pi'} L_\pi(\pi') = -\mathbb{E}_{s,a \sim D} \left[ \log \pi'(a|s) \exp \left( \beta \hat{A}^{k+1}(s, a) \right) \right]$$
$$\text{where } \hat{A}^{k+1}(s, a) = \hat{Q}^{k+1}(s, a) - \hat{V}^{k+1}(s). \tag{11}$$

Eq. 11 updates the policy $\pi$ to amounts of weighted maximum likelihood which are computed by re-weighting state-action samples in $D$ with estimated advantage $\hat{A}^{k+1}$. As discussed in the AWAC (Nair et al., 2020), this method avoids explicitly estimating the behaviour policy and its resulted sampling errors which is an import issue in the offline RL setting (Kumar et al., 2020).

**Implicit policy constraints.** We adopt the advantage weighted policy updates which imposes an implicit KL divergence constraints between $\pi$ and $\pi_\beta$. This policy constraint is necessary to guarantee that the next state $s'$ in Equation 7 can be safely generated through policy $\pi$. As derived in Nair et al. (2020) (Appendix A), the Eq. 11 is an parametric solution of the following problem:

$$\max_{\pi'} \mathbb{E}_{a \sim \pi'(\cdot|s)}[\hat{A}^{k+1}(s, a)] \quad s.t. \ D_{\text{KL}}(\pi'(\cdot|s) \| \pi_\beta(\cdot|s)) \leq \epsilon, \quad \int_a \pi'(a|s) da = 1.$$

Note that $D_{\text{KL}}(\pi' \| \pi_\beta)$ is an reserve KL divergence with respect to $\pi'$, which is mode-seeking ((Shlens, 2014)). When treated as Lagrangian it forces $\pi'$ allocate its probability mass to the maximum likelihood supports of $\pi_\beta$, re-weighted by the estimated advantage. In other words, for the space of $A$ where $\pi_\beta(\cdot|s)$ has no samples, $\pi'(\cdot|s)$ has almost zero probability mass too.

**Bonus of Exploration on Near States.** As suggested by remarks in Section 3.1, in practice allowing the policy explore the predicated next states transition ($s \sim D$) following $a \sim \pi'(\cdot|s)$) leads to better test performance. With this kind of exploration, the policy is updated as follows.

$$\pi \leftarrow \arg\min_{\pi'} L_\pi^+(\pi') = L_\pi(\pi') - \lambda \mathbb{E}_{s \sim D, a \sim \pi'(s), s' \sim \hat{P}(s,a)} \left[ r(s, a) + \hat{V}^{k+1}(s') \right] \tag{12}$$

The second term is an approximation to $E_{s \sim d_\pi(s)}[V^\pi(s)]$, while the first term is the approximation of $E_{s \sim d_u(s)}[V^\pi(s)]$. While the choice of $\lambda$ is ultimately just a hyper-parameter, we balance between optimistic policy optimization (in maximizing V) and constrained policy update (the first term) by adjusting $\lambda$.

|  |  | AWAC | CQL | COMBO | IQL | TD3-BC | PBRL | CSVE (ours) |
|---|---|---|---|---|---|---|---|---|
| Random | HalfCheetah | 12.1 | 17.5 ± 1.5 | 38.8 | - | 11.0 ± 1.1 | 13.1 ± 1.2 | 26.7 ± 2.0 |
|  | Hopper | 8.7 | 7.9 ± 0.4 | 17.9 | - | 8.5 ± 0.6 | 31.6 ± 0.3 | 27.0 ± 8.5 |
|  | Walker2D | 2.2 | 5.1 ± 1.3 | 7.0 | - | 1.6 ± 1.7 | 8.8 ± 6.3 | 6.1 ± 0.8 |
| Medium | HalfCheetah | 50.0 | 47.0 ± 0.5 | 54.2 | 47.4 | 48.3 ± 0.3 | 58.2 ± 1.5 | 48.6 ± 0.0 |
|  | Hopper | 96.0 | 53.0 ± 28.5 | 94.9 | 66.3 | 59.3 ± 4.2 | 81.6 ± 14.5 | 99.4 ± 5.3 |
|  | Walker2D | 89.1 | 73.3 ± 17.7 | 75.5 | 78.3 | 83.7 ± 2.1 | 90.3 ± 1.2 | 82.5 ± 1.5 |
| Medium Replay | HalfCheetah | 43.4 | 45.5 ± 0.7 | 55.1 | 44.2 | 44.6 ± 0.5 | 49.5 ± 0.8 | 54.8 ± 0.8 |
|  | Hopper | 97.8 | 88.7 ± 12.9 | 73.1 | 94.7 | 60.9 ± 18.8 | 100.7 ± 0.4 | 91.7 ± 0.3 |
|  | Walker2D | 80.0 | 81.8 ± 2.7 | 56.0 | 73.9 | 81.8 ± 5.5 | 86.2 ± 3.4 | 78.5 ± 1.8 |
| Medium Expert | HalfCheetah | 62.8 | 75.6 ± 25.7 | 90.0 | 86.7 | 90.7 ± 4.3 | 93.6 ± 2.3 | 93.1 ± 0.3 |
|  | Hopper | 87.2 | 105.6 ± 12.9 | 111.1 | 91.5 | 98.0 ± 9.4 | 111.2 ± 0.7 | 95.2 ± 3.8 |
|  | Walker2D | 73.7 | 107.9 ± 1.6 | 96.1 | 109.6 | 110.1 ± 0.5 | 109.8 ± 0.2 | 109.0 ± 0.1 |
| Expert | HalfCheetah | 10.3 | 96.3 ± 1.3 | - | - | 96.7 ± 1.1 | 96.2 ± 2.3 | 93.8 ± 0.1 |
|  | Hopper | 111.6 | 96.5 ± 28.0 | - | - | 107.8 ± 7 | 110.4 ± 0.3 | 111.2 ± 0.6 |
|  | Walker2D | 110.6 | 108.5 ± 0.5 | - | - | -110.2 ± 0.3 | 108.8 ± 0.2 | 108.5 ± 0.0 |

Table 1: Performance comparison on Gym control tasks. The results of CSVE is over three seeds and we reimplement AWAC using d3rlpy. Results of IQL, TD3-BC, and PBRL are from their original paper ( Table 1 in Kostrikov et al. (2021b), Table C.3 in Fujimoto & Gu (2021), and Table 1 in Bai et al. (2021) respectively. Results of COMBO are from the reproduction result in Rigter et al. (2022) (Table 1), given that the original paper report their results on v0 dataset. For the same reason, results of CQL are from Bai et al. (2021).)

## 5 EXPERIMENTS

The primary goal of this section is to investigate whether the proposed tighter conservative value estimation leads to performance improvement. Besides, we would like to ascertain when further exploration has benefits and how well CSVE performs compared with SOTA algorithms.

We evaluate our method on classical continuous control tasks of Gym(Brockman et al., 2016) and Adroit(Rajeswaran et al., 2017) in the standard D4RL (Fu et al. (2020)) benchmark. The Gym control tasks include HalfCHeetah, Hopper and Walker2D, each with 5 datasets collected by following different types of policies (random, medium, medium-replay, medium-expert, and expert). The Adroid tasks include Pen, Hammer, Door and Relocate, each with 3 dataset collected by different policies (human, cloned, and expert).

Our method, namely CSVE, and the compared baselines are CQL(Kumar et al., 2020), COMBO(Yu et al., 2021), AWACNair et al. (2020), PBRL(Bai et al., 2021) and other SOTA algorithms TD3-BC(Fujimoto & Gu, 2021), UWAC(Wu et al., 2021), IQL(Kostrikov et al., 2021b), BEAR(Kumar et al., 2019)) whose performance results are public or have high-quality open implementations. CQL which estimates the conservative Q values on state-action pairs rather than states, is the direct comparing method to ours. COMBO also lower bounds the estimated V function. AWAC is one special case of our Eq. 3 when $d = d_u$. PBRL is a very strong performant in offline RL, but is quite costly on computation since it uses the ensemble of hundreds of sub-models.

### 5.1 OVERALL PERFORMANCE

We first test on the Gym control tasks. We train our methods for 1 million steps and report the final evaluation performance. The overall results are shown in Table 1. Compared to CQL, our method has better performance on 11 of 15 tasks and similar performance on others. In particular, our method shows consistent advantage on the datasets that generated by following random or sub-optimal policies (random and medium). Compared to AWAC, our method has better performance on 9 of 15 tasks and comparable performance on others, which demonstrates the effect of our further exploration beyond cloning the behaviour policy. In particular, our method shows an obvious

|  |  | BC | BEAR | UWAC | CQL | IQL | PBRL | CSVE (ours) |
|---|---|---|---|---|---|---|---|---|
| **Human** | Pen | 34.4 | -1.0 | $10.1 \pm 3.2$ | 37.5 | 71.5 | $35.4 \pm 3.3$ | $105.0 \pm 6.1$ |
|  | Hammer | 1.5 | 0.3 | $1.2 \pm 0.7$ | 4.4 | 1.4 | $0.4 \pm 0.3$ | $3.2 \pm 2.9$ |
|  | Door | 0.5 | $-0.3$ | $0.4 \pm 0.2$ | 9.9 | 4.3 | $0.1 \pm 0.0$ | $3.1 \pm 2.7$ |
|  | Relocate | 0.0 | -0.3 | $0.0 \pm 0.0$ | 0.2 | 0.1 | $0.0 \pm 0.0$ | $0.1 \pm 0.0$ |
| **Cloned** | Pen | 56.9 | 26.5 | $23.0 \pm 6.9$ | 39.2 | 37.3 | $74.9 \pm 9.8$ | $55.2 \pm 6.1$ |
|  | Hammer | 0.8 | 0.3 | $0.4 \pm 0.0$ | 2.1 | 2.1 | $0.8 \pm 0.5$ | $0.5 \pm 0.2$ |
|  | Door | -0.1 | $-0.1$ | $0.0 \pm 0.0$ | 0.4 | 1.6 | $4.6 \pm 4.8$ | $1.3 \pm 1.1$ |
|  | Relocate | -0.1 | -0.3 | $-0.3 \pm -0.2$ | -0.1 | 0.0 | $-0.1 \pm 0.0$ | $-0.3 \pm 0.0$ |
| **Expert** | Pen | 85.1 | 105.9 | $98.2 \pm 9.1$ | 107.0 | - | $135.7 \pm 3.4$ | $142.9 \pm 10.6$ |
|  | Hammer | 125.6 | 127.3 | $107.7 \pm 21.7$ | 86.7 | - | $127.5 \pm 0.2$ | $126.5 \pm 0.4$ |
|  | Door | 34.9 | 103.4 | $104.7 \pm 0.4$ | 101.5 | - | $95.7 \pm 12.2$ | $104.3 \pm 0.9$ |
|  | Relocate | 101.3 | 98.6 | $105.5 \pm 3.2$ | 95.0 | - | $84.5 \pm 12.2$ | $103.0 \pm 1.1$ |

Table 2: Performance comparison on Adroit tasks. The results of CSVE are over three seeds. Results of IQL are from Table 3 in Kostrikov et al. (2021b) and results of other algorithm are from Table 4 in Bai et al. (2021).

advantage in extrating the best policy on data of mixed policy (Medium Expert) while AWAC can not. Compared to COMBO, our method has better performance on 6 out 12 tasks and comparable performance or slightly worse on others, which demonstrates the effect of our better bounds on V. In particular, our method shows an obvious advantage in extrating the best policy on medium and medium-expert tasks. In 9 tasks evaluated, our method gets higher score than IQL in 7 of them, and has similar performance in the other tasks. Finally, our method performs close to PBRL, even PBRL has almost orders of more model capacity and computation cost.

We now evaluate our method on the Adroit tasks. For CSVE, we report the final evaluation results after training in 0.1 million steps. The full results are reported in Table2. Copared to IQL, our method performs better in 8 out of 12 tasks, and performs similarly in the other 4 tasks. For the expert datasets, all methods including simple BC (behaviour cloning) can perform well, among which ours is the most competitive on all four tasks. For human and cloned datasets, almost all methods fail to learn effective policies on three tasks except the Pen task. For the Pen task, CSVE is the only one that succeeds to learn a good policy on the human dataset, while it can learn a medium policy on the cloned dataset as BC and PBRL.

## 5.2 Sensitiveness of hyper-parameters

We anaylyze hyper-parameter $\beta$, which trades off between behaviour cloning and policy optimization. For smaller values, the objective behaves similarly to behavior cloning (weights are close for all actions), while for larger values, it attempts to recover the maximum of the Q-function. To quantitatively analyze its effect, we test different $\beta$ from $\{0.1, 3, 10\}$ in mujoco tasks with the medium-type datasets, whose results are shown in Fig. 1. We can see that $\lambda$ has effect on the policy performance during training. Empirically, we found out that in general, $\beta = 3.0$ is suitable for such medium type datasets. Besides, in practice, by default we use $\beta = 3.0$ for random and medium task while 0.1 for medium-replay, medium-expert and expert datasets.

## 6 Related work

Offline RL (Fujimoto et al., 2019; Levine et al., 2020) aims to learn a reasonable policy from a static dataset collected by arbitrary policies, without further interactions with the environment. Compared to interactive RL, offline RL suffers two critical inherent issues, i.e., the distribution drift introduced by off-policy learning and the out-of-distribution extrapolation in value estimation (Ostrovski et al., 2021; Levine et al., 2020). The common mind of offline RL algorithms is to incorporate conservatism or regularization into the online RL algorithms. Here we briefly review the prior work with a comparison to ours.

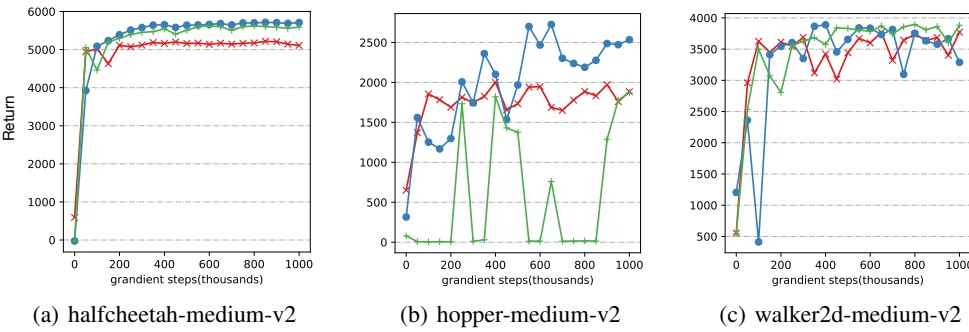

Figure 1: The effect of $\beta$ in medium tasks.

**Conservative value estimation:** Prior offline RL algorithms regularize the learning policy close to the data or explicitly estimated behaviour policy) and penalize the exploration to the out-of-distribution region, via distribution correction estimation (Dai et al., 2020; Yang et al., 2020), policy constraints with support matching (Wu et al., 2019) and distributional matching Fujimoto et al. (2019); Kumar et al. (2019), applying policy divergence based penalty on Q-functions (Kostrikov et al., 2021a; Wang et al., 2020) or uncertainty-based penalty (Agarwal et al., 2020) on Q-functions and conservative Q-function estimation (Kumar et al., 2020). Besides, model-based algorithms (Yu et al., 2020) directly estimate dynamics uncertainty and translated it into reward penalty. Different from these prior work that imposes conservatism on state-action pairs or actions, ours directly does such conservative estimation on states and requires no explicit uncertainty quantification.

With learned conservative value estimation, an offline policy can be learned via implicit derivation from a state-action joint distribution or in Q-Learning and actor-critic framework. In this paper, our implementation adopts the method proposed in AWAC (Nair et al., 2020; Peng et al., 2019a).

**Model-based algorithms:** Model-based offline RL learns the dynamics model from the static dataset and uses it to quantify uncertainty (Yu et al., 2020), data augmentention (Yu et al., 2021) with roll-outs, or planning (Kidambi et al., 2020; Chen et al., 2021). Such methods typically rely on wide data coverage when planning and data augmentation with roll-outs, and low model estimation error when estimating uncertainty, which is often difficult to satisfy in reality and leads to policy instability. Instead, we use the model to sample the next-step states only reachable from data, which has no such strict requirements on data coverage or model bias.

**Theoretical results:** Our theoretical results are derived from conservative Q-value estimation (CQL) and safe policy improvement (Laroche et al., 2019). Besides, COMBO (Yu et al., 2021) gives a result of conservative but tighter value estimation than CQL, when dataset is augmented with model-based roll-outs. Compared to our result, COMBO's lower bounds additionally assume same initial state distribution which may not always satisfy in continuous control.

## 7 DISCUSSION

In this paper, we propose a new approach for offline RL based on conservative value estimation on states and discussed how the theoretical results could lead to the new RL algorithms. In particular, we developed a practical actor-critic algorithm, in which the critic does conservative state value estimation by incorporating the penalty of the model predictive next-states into Bellman iterations, and the actor does the advantage weighted policy updates with a bonus of exploring states with conservative values. Experimental evaluation shows that our method performs better than alternative methods based on conservative Q-function estimation and is competitive among the SOTA methods, confirming our theoretical analysis well. Moving forward, we hope to explore the design of more powerful algorithms based on conservative state value estimation.

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

# A  PROOFS

We first redefine notation for clarity and then provide the proofs of the results in the main paper.

**Notation**. Let $k \in N$ denote an iteration of policy evaluation(in Section 3.2). $V^k$ denotes the true, tabular (or functional) V-function iterate in the MDP, without any correction. $\hat{V}^k$ denotes the approximate tabular (or functional) V-function iterate.

The empirical Bellman operator can be expressed as follows:

$$(\hat{\mathcal{B}}^\pi \hat{V}^k)(s) = E_{a\sim\pi(a|s)}\hat{r}(s,a) + \gamma \sum_{s'} E_{a\sim\pi(a|s)}\hat{P}(s'|s,a)[\hat{V}^k(s')] \tag{13}$$

where $\hat{r}(s,a)$ is the empirical average reward obtained in the dataset when performing action $a$ at state $s$. The true Bellman operator can be expressed as follows:

$$(\mathcal{B}^\pi V^k)(s) = E_{a\sim\pi(a|s)}r(s,a) + \gamma \sum_{s'} E_{a\sim\pi(a|s)}P(s'|s,a)[V^k(s')] \tag{14}$$

Now we first prove that the iteration in Eq.3 has a fixed point. Assume state value function is lower bounded, i.e., $V(s) \geq C \; \forall s \in S$, then Eq.3 can always be solved with Eq.4. Thus, we only need to investigate the iteration in Eq.4.

Denote the iteration as a function operator $\mathcal{T}^\pi$ on $V$. Suppose $\operatorname{supp} d \subseteq \operatorname{supp} d_u$, then the operator $\mathcal{T}^\pi$ is a $\gamma$-contraction in $L_\infty$ norm where $\gamma$ is the discounting factor.

**Proof of Lemma 1:** Let $V$ and $V'$ are any two state value functions with the same support, i.e., $\operatorname{supp}V = \operatorname{supp}V'$.

$$|(\mathcal{T}^\pi V - \mathcal{T}^\pi V')(s)| = \left| (\hat{\mathcal{B}}^\pi V(s) - \alpha[\frac{d_{(s)}}{d_u(s)} - 1]) - (\hat{\mathcal{B}}^\pi V'(s) - \alpha[\frac{d_{(s)}}{d_u(s)} - 1]) \right|$$

$$= \left| \hat{\mathcal{B}}^\pi V(s) - \hat{\mathcal{B}}^\pi V'(s) \right|$$

$$= |(E_{a\sim\pi(a|s)}\hat{r}(s,a) + \gamma E_{a\sim\pi(a|s)}\sum_{s'}\hat{P}(s'|s,a)V(s'))$$

$$- (E_{a\sim\pi(a|s)}\hat{r}(s,a) + \gamma E_{a\sim\pi(a|s)}\sum_{s'}\hat{P}(s'|s,a)V'(s'))|$$

$$= \gamma \left| E_{a\sim\pi(a|s)}\sum_{s'}\hat{P}(s'|s,a)[V(s') - V'(s')] \right|$$

$$||\mathcal{T}^\pi V - \mathcal{T}^\pi V'||_\infty = \max_s |(\mathcal{T}^\pi V - \mathcal{T}^\pi V')(s)|$$

$$= \max_s \gamma \left| E_{a\sim\pi(a|s)}\sum_{s'}\hat{P}(s'|s,a)[V(s') - V'(s')] \right|$$

$$\leq \gamma E_{a\sim\pi(a|s)}\sum_{s'}\hat{P}(s'|s,a)\max_{s''}|V(s'') - V'(s'')|$$

$$= \gamma \max_{s''}|V(s'') - V'(s'')|$$

$$= \gamma||(V - V')||_\infty$$

$\square$

We present the bound on using empirical Bellman operator compared to the true Bellman operator. Following previous work Kumar et al. (2020), we make the following assumptions that: $P^\pi$ is the transition matrix coupled with policy, specifically, $P^\pi V(s) = E_{a'\sim\pi(a'|s'),s'\sim P(s'|s,a')}[V(s')]$

**Assumption 1.** $\forall s, a \in \mathcal{M}$, the following relationships hold with at least $(1 - \delta)$ ($\delta \in (0,1)$) probability,

$$|r - r(s,a)| \leq \frac{C_{r,\delta}}{\sqrt{|D(s,a)|}}, ||\hat{P}(s'|s,a) - P(s'|s,a)||_1 \leq \frac{C_{P,\delta}}{\sqrt{|D(s,a)|}} \tag{15}$$

Under this assumption, the absolute difference between the empirical Bellman operator and the actual one can be calculated as follows:

$$|(\hat{\mathcal{B}}^\pi)\hat{V}^k - (\mathcal{B}^\pi)\hat{V}^k)| = E_{a\sim\pi(a|s)}|r - r(s,a) + \gamma \sum_{s'} E_{a'\sim\pi(a'|s')}(\hat{P}(s'|s,a) - P(s'|s,a))[\hat{V}^k(s')]| \tag{16}$$

$$\leq E_{a\sim\pi(a|s)}|r - r(s,a)| + \gamma|\sum_{s'} E_{a'\sim\pi(a'|s')}(\hat{P}(s'|s,a') - P(s'|s,a'))[\hat{V}^k(s')]| \tag{17}$$

$$\leq E_{a\sim\pi(a|s)}\frac{C_{r,\delta} + \gamma C_{P,\delta}2R_{max}/(1-\gamma)}{\sqrt{|D(s,a)|}} \tag{18}$$

Thus, the estimation error due to sampling error can be bounded by a constant as a function of $C_{r,\delta}$ and $C_{t,\delta}$. We define this constant as $C_{r,T,\delta}$.

Thus we obtain:

$$\forall V, s \in D, |\hat{\mathcal{B}}^\pi V(s) - \mathcal{B}^\pi V(s)| \leq E_{a\sim\pi(a|s)}\frac{C_{r,t,\delta}}{(1-\gamma)\sqrt{|D(s,a)|}} \tag{19}$$

Next we provide an important lemma.

**Lemma 2.** *(Interpolation Lemma) For any $f \in [0,1]$, and any given distribution $\rho(s)$, let $d_f$ be an $f$-interpolation of $\rho$ and $D$, i.e.,$d_f(s) := fd(s) + (1-f)\rho(s)$, let $v(\rho,f) := E_{s\sim\rho(s)}[\frac{\rho(s)-d(s)}{d_f(s)}]$, then $v(\rho,f)$ satisfies $v(\rho,f) \geq 0$.*

The proof can be found in Yu et al. (2021). By setting $f$ as 1, we have $E_{s\sim\rho(s)}[\frac{\rho(s)-d(s)}{d(s)}] > 0$.

**Proof of Theorem 1:** The V function of approximate dynamic programming in iteration $k$ can be obtained as:

$$\hat{V}^{k+1}(s) = \hat{\mathcal{B}}^\pi\hat{V}^k(s) - \alpha[\frac{d(s)}{d_u(s)} - 1] \forall s, k \tag{20}$$

The fixed point:

$$\hat{V}^\pi(s) = \hat{\mathcal{B}}^\pi\hat{V}^\pi(s) - \alpha[\frac{d(s)}{d_u(s)} - 1] \leq \mathcal{B}^\pi\hat{V}^\pi(s) + E_{a\sim\pi(a|s)}\frac{C_{r,t,\delta}R_{max}}{(1-\gamma)\sqrt{|D(s,a)|}} - \alpha[\frac{d(s)}{d_u(s)} - 1] \tag{21}$$

Thus we obtain:

$$\hat{V}^\pi(s) \leq V^\pi(s) + (I - \gamma P^\pi)^{-1}E_{a\sim\pi(a|s)}\frac{C_{r,t,\delta}R_{max}}{(1-\gamma)\sqrt{|D(s,a)|}} - \alpha(I - \gamma P^\pi)^{-1}[\frac{d(s)}{d_u(s)} - 1] \tag{22}$$

, where $P^\pi$ is the transition matrix coupled with the policy $\pi$ and $P^\pi V(s) = E_{a'\sim\pi(a'|s')s'\sim P(s'|s,a')}[V(s')]$.

Then the expectation of $V^\pi(s)$ under distribution $d(s)$ satisfies:

$$E_{s\sim d(s)}\hat{V}^\pi(s) \leq E_{s\sim d(s)}(V^\pi(s)) + E_{s\sim d(s)}(I - \gamma P^\pi)^{-1}E_{a\sim\pi(a|s)}\frac{C_{r,t,\delta}R_{max}}{(1-\gamma)\sqrt{|D(s,a)|}}$$
$$-\alpha \underbrace{E_{s\sim d(s)}(I - \gamma P^\pi)^{-1}[\frac{d(s)}{d_u(s)} - 1])}_{>0} \tag{23}$$

When $\alpha \geq \frac{E_{s\sim d(s)}E_{a\sim\pi(a|s)}\frac{C_{r,t,\delta}R_{max}}{(1-\gamma)\sqrt{|D(s,a)|}}}{E_{s\sim d(s)}[\frac{d(s)}{d_u(s)}-1])}$, $E_{s\sim d(s)}\hat{V}^\pi(s) \leq E_{s\sim d(s)}(V^\pi(s))$. $\qquad\square$

**Proof of Theorem 2:** The expectation of $V^\pi(s)$ under distribution $d(s)$ satisfies:

$$E_{s \sim d_u(s)} \hat{V}^\pi(s) \leq E_{s \sim d_u(s)}(V^\pi(s)) + E_{s \sim d_u(s)}(I - \gamma P^\pi)^{-1} E_{a \sim \pi(a|s)} \frac{C_{r,t,\delta} R_{max}}{(1-\gamma)\sqrt{|D(s,a)|}}$$
$$- \alpha E_{s \sim d_u(s)}(I - \gamma P^\pi)^{-1}[\frac{d(s)}{d_u(s)} - 1]) \tag{24}$$

Noticed that the last term:

$$\sum_{s \sim d_u(s)} (\frac{d_f(s)}{d_u(s)} - 1) = \sum_s d_u(s)(\frac{d_f(s)}{d_u(s)} - 1) = \sum_s d_f(s) - \sum_s d_u(s) = 0 \tag{25}$$

We obtain that:

$$E_{s \sim d_u(s)} \hat{V}^\pi(s) \leq E_{s \sim d_u(s)}(V^\pi(s)) + E_{s \sim d_u(s)}(I - \gamma P^\pi)^{-1} E_{a \sim \pi(a|s)} \frac{C_{r,t,\delta} R_{max}}{(1-\gamma)\sqrt{|D(s,a)|}} \tag{26}$$

$\square$

**Proof of Theorem 3:** Recall that the expression of the V-function iterate is given by:

$$\hat{V}^{k+1}(s) = \mathcal{B}^{\pi^k} \hat{V}^k(s) - \alpha[\frac{d(s)}{d_u(s)} - 1] \forall s, k \tag{27}$$

Now the expectation of $V^\pi(s)$ under distribution $d_u(s)$ is given by:

$$E_{s \sim d_u(s)} \hat{V}^{k+1}(s) = E_{s \sim d_u(s)} \left[ \mathcal{B}^{\pi^k} \hat{V}^k(s) - \alpha[\frac{d(s)}{d_u(s)} - 1] \right] = E_{s \sim d_u(s)} \mathcal{B}^{\pi^k} \hat{V}^k(s) \tag{28}$$

The expectation of $V^\pi(s)$ under distribution $d(s)$ is given by:

$$E_{s \sim d(s)} \hat{V}^{k+1}(s) = E_{s \sim d(s)} \mathcal{B}^{\pi^k} \hat{V}^k(s) - \alpha[\frac{d(s)}{d_u(s)} - 1] = E_{s \sim d(s)} \mathcal{B}^{\pi^k} \hat{V}^k(s) - \alpha E_{s \sim d(s)}[\frac{d(s)}{d_u(s)} - 1] \tag{29}$$

Thus we can show that:

$$E_{s \sim d_u(s)} \hat{V}^{k+1}(s) - E_{s \sim d(s)} \hat{V}^{k+1}(s) = E_{s \sim d_u(s)} \mathcal{B}^{\pi^k} \hat{V}^k(s) - E_{s \sim d(s)} \mathcal{B}^{\pi^k} \hat{V}^k(s) + \alpha E_{s \sim d(s)}[\frac{d(s)}{d_u(s)} - 1]$$
$$= E_{s \sim d_u(s)} V^{k+1}(s) - E_{s \sim d(s)} V^{k+1}(s) - E_{s \sim d(s)}[\mathcal{B}^{\pi^k}(\hat{V}^k - V^k)]$$
$$+ E_{s \sim d_u(s)}[\mathcal{B}^{\pi^k}(\hat{V}^k - V^k)] + \alpha E_{s \sim d(s)}[\frac{d(s)}{d_u(s)} - 1] \tag{30}$$

By choosing $\alpha$:

$$\alpha > \frac{E_{s \sim d(s)}[\mathcal{B}^{\pi^k}(\hat{V}^k - V^k)] - E_{s \sim d_u(s)}[\mathcal{B}^{\pi^k}(\hat{V}^k - V^k)]}{E_{s \sim d(s)}[\frac{d(s)}{d_u(s)} - 1]} \tag{31}$$

We have $E_{s \sim d_u(s)} \hat{V}^{k+1}(s) - E_{s \sim d(s)} \hat{V}^{k+1}(s) > E_{s \sim d_u(s)} V^{k+1}(s) - E_{s \sim d(s)} V^{k+1}(s)$ hold. $\square$

**Proof of Theorem 4:** $\hat{V}$ is obtained by solving the recursive Bellman fixed point equation in the empirical MDP, with an altered reward, $r(s, a) - \alpha[\frac{d(s)}{d_u(s)} - 1]$, hence the optimal policy $\pi^*(a|s)$ obtained by optimizing the value under Eq. 4. $\square$

**Proof of Theorem 5:** The proof of this statement is divided into two parts. We first relates the return of $\pi^*$ in the empirical MDP $\hat{M}$ with the return of $\pi_\beta$, we can get:

$$J(\pi^*, \hat{M}) - \alpha \frac{1}{1-\gamma} \mathbb{E}_{s \sim d_{\hat{M}}^{\pi^*}(s)}[\frac{d(s)}{d_u(s)} - 1] \geq J(\pi_\beta, \hat{M}) - 0 = J(\pi_\beta, \hat{M}) \tag{32}$$

The next step is to bound the difference between $J(\pi_\beta, \hat{M})$ and $J(\pi_\beta, M)$ and the difference between $J(\pi^*, \hat{M})$ and $J(\pi^*, M)$. We quote a useful lemma from Kumar et al. (2020) (Lemma D.4.1):

**Lemma 3.** *For any MDP $M$, an empirical MDP $\hat{M}$ generated by sampling actions according to the behavior policy $\pi_\beta$ and a given policy $\pi$,*

$$|J(\pi, \hat{M}) - J(\pi, M)| \leq \left(\frac{C_{r,\delta}}{1-\gamma} + \frac{\gamma R_{max} C_{T,\delta}}{(1-\gamma)^2}\right) \mathbb{E}_{s \sim d_{\hat{M}}^{\pi^*}(s)} \left[\frac{\sqrt{|\mathcal{A}|}}{\sqrt{|\mathcal{D}(s)|}} \sqrt{E_{a \sim \pi(a|s)}\left(\frac{\pi(a|s)}{\pi_\beta(a|s)}\right)}\right] \quad (33)$$

Setting $\pi$ in the above lemma as $\pi_\beta$, we get:

$$|J(\pi_\beta, \hat{M}) - J(\pi_\beta, M)| \leq \left(\frac{C_{r,\delta}}{1-\gamma} + \frac{\gamma R_{max} C_{T,\delta}}{(1-\gamma)^2}\right) \mathbb{E}_{s \sim d_{\hat{M}}^{\pi^*}(s)} \left[\frac{\sqrt{|\mathcal{A}|}}{\sqrt{|\mathcal{D}(s)|}} \sqrt{E_{a \sim \pi^*(a|s)}\left(\frac{\pi^*(a|s)}{\pi_\beta(a|s)}\right)}\right]$$
$$(34)$$

, given that $\sqrt{E_{a \sim \pi^*(a|s)}\left[\frac{\pi^*(a|s)}{\pi_\beta(a|s)}\right]}$ is a pointwise upper bound of $\sqrt{E_{a \sim \pi_\beta(a|s)}\left[\frac{\pi_\beta(a|s)}{\pi_\beta(a|s)}\right]}$(Kumar et al. (2020)). Thus we get,

$$J(\pi^*, \hat{M}) \geq J(\pi_\beta, \hat{M}) - 2\left(\frac{C_{r,\delta}}{1-\gamma} + \frac{\gamma R_{max} C_{T,\delta}}{(1-\gamma)^2}\right) \mathbb{E}_{s \sim d_{\hat{M}}^{\pi^*}(s)} \left[\frac{\sqrt{|\mathcal{A}|}}{\sqrt{|\mathcal{D}(s)|}} \sqrt{E_{a \sim \pi^*(a|s)}\left(\frac{\pi^*(a|s)}{\pi_\beta(a|s)}\right)}\right]$$
$$+ \alpha \frac{1}{1-\gamma} \mathbb{E}_{s \sim d_{\hat{M}}^{\pi}(s)}\left[\frac{d(s)}{d_u(s)} - 1\right]$$
$$(35)$$

, which completes the proof. $\square$

Here, the second term is sampling error which occurs due to mismatch of $\hat{M}$ and $M$; the third term denotes the increase in policy performance due to CSVE in $\hat{M}$. Note that when the first term is small, the smaller value of $\alpha$ are able to provide an improvement compared to the behavior policy.

## B  CSVE ALGORITHM

Now we put all in section 4 together and describe the practical deep offline reinforcement learning algorithm. In particular, the dynamic model model, value functions and policy are all parameterized with deep neural networks and trained via stochastic gradient decent methods. The pseudo code is given in Alg. 1.

---

**Algorithm 1:** CSVE Algorithm

**Input** : Data $D = \{(s, a, r, s')\}$
**Parameters:** $Q_\theta, V_\psi, \pi_\phi, Q_{\bar{\theta}}, M_\nu$
**Hyperparameters:** $\alpha, \lambda$, learning rates $\eta_\theta, \eta_\psi, \eta_\phi, \omega$
**begin**
    // Train transition model with the static dataset $D$
1    $M_\nu \leftarrow train(D)$;
    // Train the conservative value and policy functions
2    Initialize function parameters $\theta_0, \psi_0, \phi_0, \bar{\theta}_0 = \theta_0$;
3    **foreach** *step* $k = 1 \rightarrow N$ **do**
4        $\psi_k \leftarrow \psi_{k-1} - \eta_\psi \nabla_\psi L_V^\pi(V_\psi; \overline{\hat{Q}_{\theta_k}})$;
5        $\theta_k \leftarrow \theta_{k-1} - \eta_\theta \nabla_\theta L_Q^\pi(Q_\theta; \hat{V}_{\psi_k})$;
6        $\phi_k \leftarrow \phi_{k-1} - \eta_\phi \nabla_\phi L_\pi^+(\pi_\phi)$;
7        $\bar{\theta}_k \leftarrow \omega \bar{\theta}_{k-1} + (1 - \omega)\theta_k$;

---

## C  IMPLEMENTATION DETAIL

We implement our method based on an offline deep reinforcement learning library d3rlpy (Seno & Imai, 2021). The code is available at https://github.com/iclr20234089/code4098.

The detailed hyper-parameters are provided in Table 3

| Hyper-parameters | Value and description |
|---|---|
| B | 5, number of ensembles in dynamics model |
| $\alpha$ | 10, to control the penalty of out-of-distribution states |
| $\tau$ | 10, budget parameter in Eq. 10 |
| $\beta$ | In Gym domain, 3 for random and medium tasks, 0.1 for the other tasks; In Adroit domain, 30 for human and cloned tasks, 0.01 for expert tasks |
| $\gamma$ | 0.99, discount factor. |
| H | 1 million for Mujoco while 0.1 million for Adroit tasks. |
| w | 0.005, target network smoothing coefficient. |
| lr of actor | 3e-4, policy learning rate |
| lr of critic | 1e-4, critic learning rate |

Table 3: Hyer-parameters

# D EXTENDED EXPERIMENTAL RESULTS

## D.1 MORE EXPERIMENTS ON HYPER-PARAMETERS EFFECT

We also investigated $\lambda$ values of $\{0.0, 0.1, 0.5, 1.0\}$ in the medium tasks. The results are shown in Fig. 4.

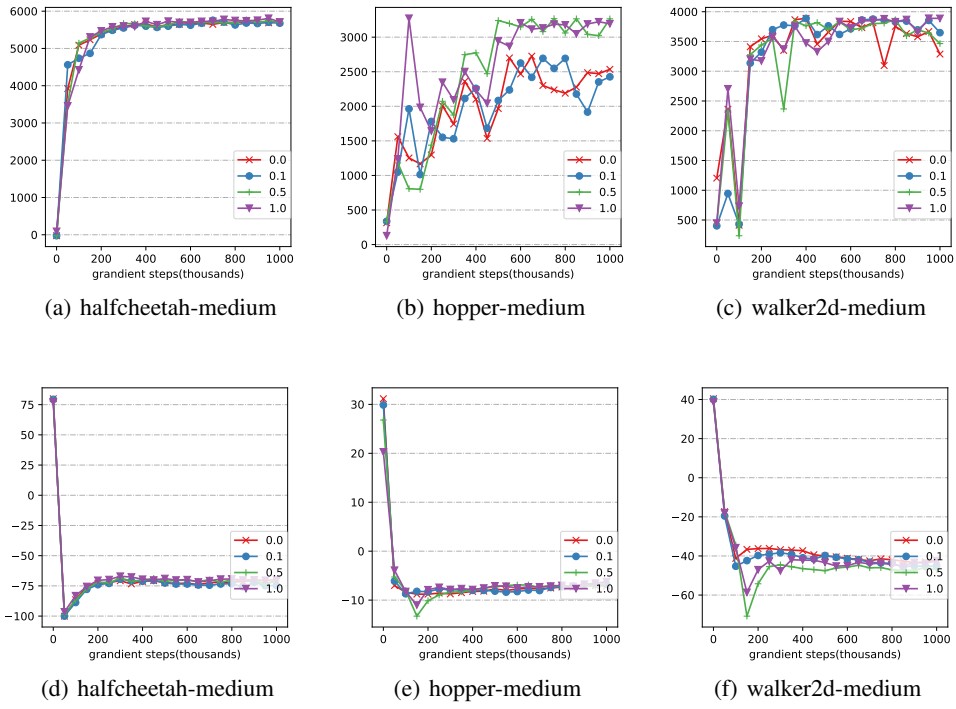

      (a) halfcheetah-medium        (b) hopper-medium        (c) walker2d-medium

      (d) halfcheetah-medium        (e) hopper-medium        (f) walker2d-medium

Figure 2: Return (upper sub-figures) and loss in Eq. 11 (bottom sub-figures) during training with different $\lambda$ values

## D.2 COMPARISON WITH PESSIMISM ON Q

We implement an ablation version of our method–penalty-Q, which directlly penalize the value of state action pairs. Specifically, we change the critic loss function into :

$$\hat{Q}^{k+1} \leftarrow \underset{Q}{\arg\min} \, L_Q^\pi(Q; \overline{\hat{Q}^k}) = \alpha \left( \mathbb{E}_{s \sim D, a' \sim \pi(\cdot|s)}[Q(s, a')] - \mathbb{E}_{s, a \sim D}[Q(s, a)] \right)$$
$$+ \mathbb{E}_{s, a, s' \sim D} \left[ \left( r(s, a) + \gamma \hat{Q}^{k+1}(s', a') - Q(s, a) \right)^2 \right] \quad (36)$$

We use the same policy extraction method and test this method on the medium-task, in which the data is collected using a medium-performed policy. In all the three tasks, the performance of the penalty-Q are worse than the the original implementation, the penalty-V counterpart. When penalty is on the state-action pair, as illustrated by our theoretical discussion, the value of the evaluated Q value tends to pointwise lower bounds the true Q value, which results in a more conservative and thus worse policy. While when we penalize V, the estimated value function only bounds the expectation of the true V function, which results in a more flexible and well-performed policy.

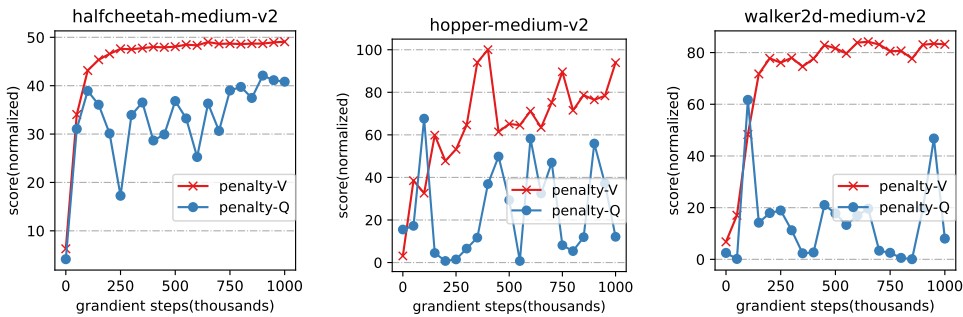

Figure 3: Permance comparison between our original implementation and the penalty-Q version.

### D.3 Relationship between model bias and final performance

As stated in the main paper, compared to normal model-based offline RL algorithms, CSVE is insensitive to model biases. To understand this quantitatively, now we investigate the effect of model biases to the performance. We use the the dynamic model's average L2 error on transition prediction as the surrogate of model biases. As shown in Fig. 4, in CSVE, the model bias has very little effect to RL performance. Particularly, for halfcheetah there is observed effect of model errors to scores, while in hopper and walker2d with increasing errors, the scores have a slight downward trend where the decrease is relatively very small.

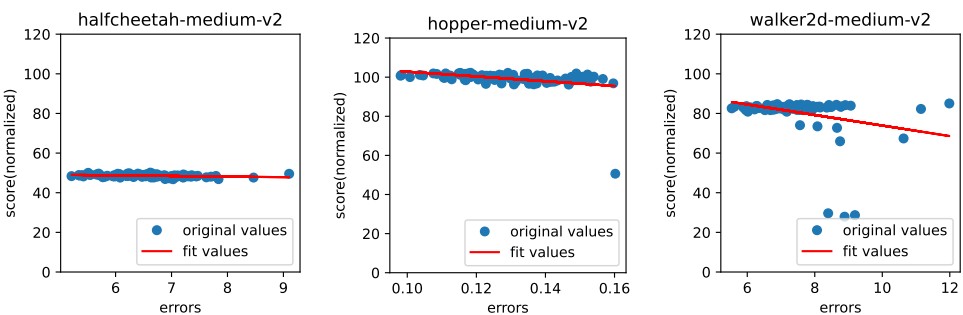

Figure 4: The relationship between score and the model biases. The correlation coefficient is respectively $-0.32$, $-0.34$, and $-0.29$.

## D.4 Reproduction of COMBO

In the main body of this paper, our results of COMBO adopt the results presented in literature (Rigter et al., 2022). Our goal here is to look into more details of COMBO's asymptotic performance evaluated during training. For comparison fairness, we adopt the official COMBO code provided by author, and rerun to evaluate with the medium dataset of D4RL mujoco v2.

Fig. 5 shows the asymptotic performance until 1000 epochs, in which the scores have been normalized with corresponding SAC performance. We found that in both hopper and walker2d, the scores show dramatic fluctuations. The average scores of last 10 epochs for halfcheetah, hopper and walker2d are 71.7, 65.3 and -0.26 in respect. Besides, we found even in D4RL v0 dataset, COMBO's behaviours are similar.

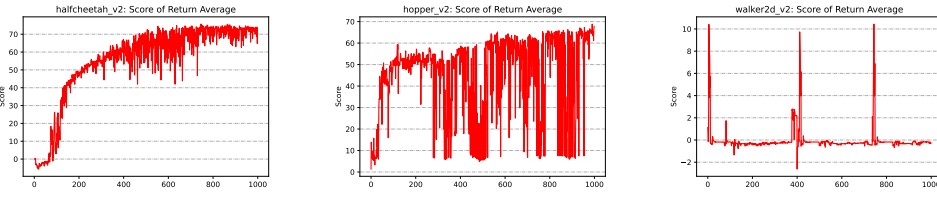

Figure 5: Return of COMBO on D4RL mujoco v2 tasks

## D.5 Effect of Exploration near Dataset Distributions

As discussed in Section 3.1 and 4.2, proper choice of exploration on the distribution ($d$) beyond data ($d_u$) should help policy improvement. The factor $\lambda$ in Eq. 12 controls the trade-off on such 'bonus' exploration and complying the data-implied behaviour policy.

Let us take the medium-replay type of datasets to analyze its effect. In the experiments, with fixed $\beta = 0.1$, we investigate $\lambda$ values of $\{0.0, 0.5, 1.0, 3.0\}$. As shown in the upper figures in Fig. 6, $\lambda$ shows obvious effect to policy performance and variances during training. In general, there is a value under which increasing $\lambda$ leads to performance improvement, while above which further increasing $\lambda$ hurts performance. For example, with $\lambda = 3.0$ in hopper-medium-replay task and walker2d-medium-replay task, the performance get worse than with smaller $\lambda$ values. The value of $\lambda$ is task-specific, and we find that its effect is highly related to the loss in Eq. 11 which can be observed by comparing bottom and upper figures in Fig. 6. Thus, in practice, we can choose proper $\lambda$ according to the above loss without online interaction.

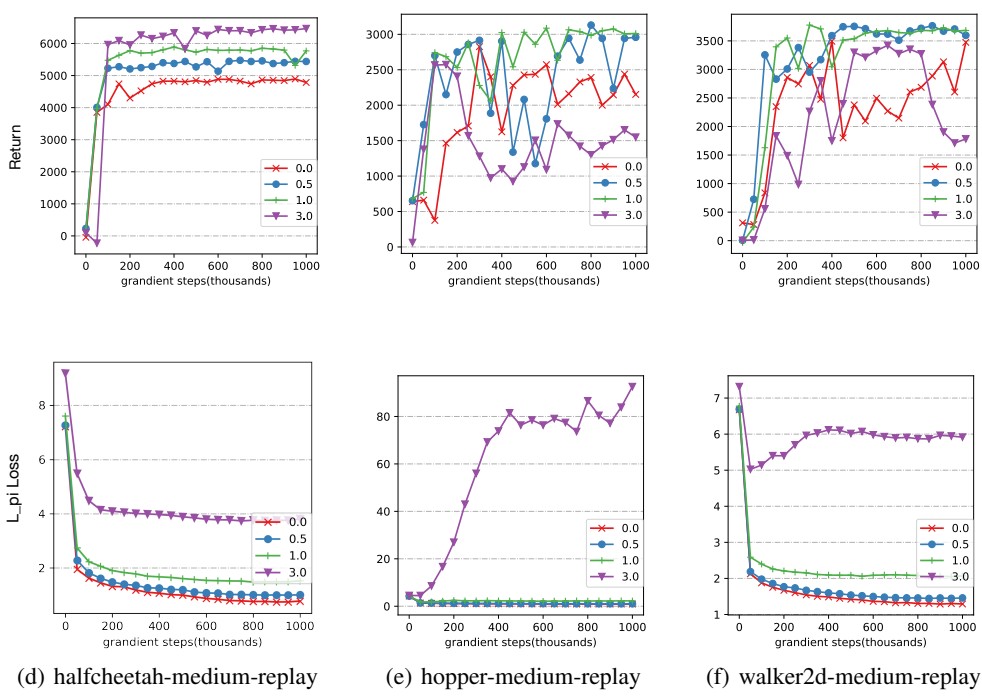

Figure 6: Effect of $\lambda$ to Return (upper figures) and $L_\pi$ loss in Eq. 11 (bottom figures)

