# OpenReview forum: "Effective Offline Reinforcement Learning via Conservative State Value Estimation"
_ICLR.cc/2023/Conference — Submitted to ICLR 2023_

### Official Review · Reviewer_zsLW · 2022-10-24

**Confidence:** 3
**Correctness:** 4
**Technical Novelty And Significance:** 3
**Empirical Novelty And Significance:** 2
**Recommendation:** 8

**Clarity, Quality, Novelty And Reproducibility:**

Both the technical and presentation quality of the paper is good, and the clarity is also OK. However, there are some typos in the paper. The algorithm suggested in the paper is also quite original.

**Strength And Weaknesses:**

Strength
- Theoretically well supported
- Quite an original idea and algorithm design
- Descent performance

Weakness
- Requires model:

When compared to other model-free methods, CSVE requires estimating dynamics as well, so it may not be very fair to compare to other model-free algorithms. Since the authors argue that the proposed method is not very sensitive to model bias, it would be also interesting to see how CSVE is affected by varying model bias.

- Not enough interpretation on why the suggested algorithm is good:

It is kind of vague in the paper why suggested algorithm works well when compared to CQL. Is it really easy to learn pessimistic V value instead of Q? Can it be shown as an example? Or is V-learning algorithm is just better than Q-learning in these domains, even in the online setting? It is hard to see the clear reason from the paper.

**Summary Of The Paper:**

This paper proposes an algorithm called CSVE, which imposes penalization on states rather than state actions used in CQL. By doing such, we may benefit from learning a better conservative value since learning a conservative Q-value over joint state-action space is more challenging. The paper provides theoretical results that are improved from that of CQL. Although it requires learning a model to learn, the experiment results suggest that CCSVE overall performs better than CQL.

**Summary Of The Review:**

Overall, I enjoyed reading this paper and I think there will be a number of readers interested in this paper. There are quite a few typos in the papers, and I would like to see more empirical results (e.g. when the model is incorrect), but overall I recommend acceptance of this paper.

---

> ### Author Response · Authors · 2022-11-19
> **Response to Reviewer zsLW**
>
> Thank you for positive reviews and constructive advice!
>
> **Concern 1: Model's effect to CSVE performance**
>
> > Fairness of CSVE compared to model-free algorithms
>
> We have added detailed comparison of methodoly and experiments to COMBO, the most related model-based offline RL algoritm, as shown in the common response. Besides, except leveraging a model the sample OOD states, CSVE is more like to model-free algorithms.
>
> > Sensitiveness of CSVE to model bias
>
> In principle, CSVE should not be sensitive to model bias, since it uses model only for sampling OOD states rather than generating rollouts to train the RL critic or actor. We have added experiments to investigate the correlation of performance and different model biases in appendix D.3.
>
> **Concern 2: Interpretation on why CSVE is better than CQL**
> Compared to CQL and its common implementation, CSVE has advantage on two aspects. On the conservative value estimation, CSVE relaxes CQL's guarantee of state-wise lower bounds to expectation lower bounds, which makes CSVE has more potential on policy improvement. On the algorithmic implementation, our method adopts model-based ood state sampling and AWR for policy extraction.

---

### Official Review · Reviewer_pQB8 · 2022-10-25

**Confidence:** 2
**Clarity, Quality, Novelty And Reproducibility:** Please see above section.
**Correctness:** 2
**Technical Novelty And Significance:** 2
**Empirical Novelty And Significance:** 2
**Recommendation:** 3

**Strength And Weaknesses:**

Strength:

1. The paper studies an interesting topic which could be practically useful to a broad range of RL applications.

2. The paper has a reasonably clear presentation.

Weaknesses:

---------
1. The basic idea is incremental to CQL. The actions can be marginalized from the CQL’s bound to get the state value bound. But the paper does not discuss any connections.

On close examination, both the theoretical result and algorithmic design are highly similar to CQL.

---------
2. The theorems are not informative. First, the theoretical results are bounding E[V(s)], but CQL provides a bound for any state-action pair. Bounding E[V(s)] does not give a guarantee for any state value estimate’s conservativeness.

Second, before showing the bound, shouldn’t the convergence be shown first? To me, it is even unclear if the proposed conservative state value Bellman update would converge or not.

---------
4. Experiments. I expect to see clear evidence to show at least three things:
1). The necessity of learning a conservative state value, rather than action-value;
2). The the effectiveness of using the proposed method to learn conservative state value, rather than other methods (for example, using CQL and use the learned policy to get a state value estimate);
3). The comparison to IQL (Offline RL with implicit q learning by Ilya et al.), which has a similar high-level idea. IQL attempts to avoid overestimation by avoiding learning action values.

**Summary Of The Paper:**

A major challenge in offline RL is the distribution mismatch between the behavior policy generating the offline data and the target policy we want to learn. Such a mismatch can result in an overestimation error in the Bellman update, which can further result in divergence. The paper proposes to learn a conservative state value estimate (in contrast to the existing CQL, which learns conservative action-value estimates). The conservative state-value estimate is learned by subtracting a term from the regular Bellman backup. The term is proportional to the density ratio of the target and behavior distribution. The main theory shows the bound of the expected state value (i.e., expectation over states) under both the offline data distribution and the target policy distribution. In algorithmic design, the conservative state value estimate is used to learn a critic in an actor-critic algorithm. Experiments on Mujoco domains are conducted to show its effectiveness.


**Summary Of The Review:**

I have limited time to review the paper. But to the best knowledge, the current weaknesses of the paper outweigh the strength. I will refer to the author's response and other reviews and adjust my score accordingly.

---

> ### Author Response · Authors · 2022-11-19
> **Response to Reviewer pQB8**
>
> Thank you very much for such constructive comments! We shall improve it in the future version, including the comparison with CQL, convergence proof, effect of bounding E[V[s]], and more experimental results.
>
> **Concern 1: Comparision with CQL**
>
> We admit that in the theory part, we closely follow the notations and main theorem framework of CQL. This is indeed our intent which lets the readers easier to follow and should be claimed in our paper. Besides our work, COMBO, we believe, follows the same tradition of CQL too.
>
> **We argue that CSVE in its concept is an alternative approach rather than simply incremental to CQL.** CQL penalizes *the OOD actions conditioned on states in the dataset $D$*, and guarantees the state-wise lower bounds of state values: $\hat{V}^\pi(s) \leq V^\pi(s)$ for all $s \in D$. CSVE penalizes *the OOD states directly*, and guarantees the lower bounds of state value expectation: $\mathbb E_{s \sim d}[\hat{V}^{\pi}(s)] \leq \mathbb E_{s \sim d}[V^{\pi}(s)]$. In the common response, we discussed in detail the theoretical and practical sense of our methods compared to CQL as well as COMBO.
>
> On the algorithm or practical implementation part, however ours is quite different from the CQL paper, except that the we borrowed the idea of adaptive $\alpha$ from the original CQL paper. Take the actor-critic algorithm as an example. For learning critic, CQL uses random policy or learned policy to do the ood action sampling, while ours use the learned policy and dynamic model to sample ood states. During learning actor, CQL uses a SAC-style method or direct argmax (for discrete actions) to do policy extraction, while ours uses AWR with an optional exploration term.
>
> > "The actions can be marginalized from the CQL’s bound to get the state value bound. But the paper does not discuss any connections."
>
> We have added the detailed discussion in the common response. As stated there, CQL's state value bound is different from CSVE.
>
>
> **Concern 2: Issues of bounding $\mathbb{E}[V[s]]$**
> As discussed in the common response, CSVE guarantees a lower bound of the $\mathbb E_{s \sim d}[V^\pi[s]]$, while COMBO get the same result on the special case where $d = \mu_0$ (see also Remark 1, section A.2 of COMBO[2]). As a comparison, CQL gets lower bounds of $V(s)$ for all $s \in D$.
>
> Bounding $\mathbb{E}[V[s]]$ would introduce a more adventurous policy, which would achieves better performance in in-distribution states and have more risk behaviors in OOD states. To deal with that limitation, we introduce a model to sample the OOD states for better conservative estimation.
>
>
> **Concern 3: Bellman convergence of CSVE**
> We will add this part to the paper revision. The main idea is to prove that the function projection in Eq.4 is a $\gamma$-contraction mapping under some mild assumptions.
>
> Under the mild assumption that state value function is lower bounded, i.e., $V(s) \geq C$ for all $s \in S$, then Eq.3 can always be solved with Eq.4. Denote the iteration in Eq.4 as the function operator $\mathcal{T}^\pi$ on $V$. Suppose $\mathtt{supp} d \subseteq \mathtt{supp} d_u$, then the operator $\mathcal{T}^\pi$ is a $\gamma$-contraction in $L_\infty$ norm where $\gamma$ is the discounting factor.
>
> **Proof:**
> Let $V$ and $V'$ are any two state value functions with the same support, i.e., $\mathtt{supp}{V} = \mathtt{supp}{V'}$.
> $$
>     \left| (\mathcal{T}^\pi V - \mathcal{T}^\pi V')(s) \right|
>     = \left|(\hat{\mathcal B^{\pi}}V(s) - \alpha[ \frac{d_{(s)}}{d_u(s)}-1 ]) - (\hat{\mathcal B^{\pi}}V'(s) - \alpha[ \frac{d_{(s)}}{d_u(s)}-1 ])\right|
>     = \left| \hat{\mathcal B^{\pi}}V(s) - \hat{\mathcal B^{\pi}}V'(s) \right|
>     = \gamma \left| E_{a \sim \pi(a|s)} \sum_{s'} \hat{P}(s'|s,a) [V(s') - V'(s')] \right|
> $$
> Then we have:
> $$
> || \mathcal{T}^\pi V - \mathcal{T}^\pi V' || _ {\infty}
>     = \max_{s} \left| (\mathcal{T}^\pi V - \mathcal{T}^\pi V')(s) \right|
>     = \max_{s} \gamma | E_{a \sim \pi(a|s)} \sum_{s'} \hat{P}(s'|s,a) [V(s') - V'(s')] |
>     \leq \gamma E_{a \sim \pi(a|s)} \sum_{s'} \hat{P}(s'|s,a) \max_{s''}|V(s'') - V'(s'')|
>     = \gamma \max_{s''} \left| V(s'') - V'(s'') \right|
>     = \gamma || (V - V')||_\infty
> $$
> When $\gamma < 1$, the contraction projection should converge to a fixed point asymptotically.
>
> **Concern 4: More experimental comparison**
> > Necessity and effectiveness of conservative state values
>
> In the revised paper, we have added experiments to evaluate performance effect of conservative state values. As our emperical stdudy in appendix D.2 shows, in our CSVE framework, adding conservatism on V rather than on Q leads to more stable and better asymtotic performane.
>
> >Comparison with IQL
>
> We compared our model with IQL on D4RL mujuco tasks v2. As shown in Table. A1 of the common response, CSVE performs better in 7 out of 9 tasks. Considering both CSVE and IQL use the AWR for policy extraction, the above performance advantage should be mainly attributed to our conservative state estimation approach.

---

### Official Review · Reviewer_KLUN · 2022-10-27

**Confidence:** 4
**Correctness:** 3
**Technical Novelty And Significance:** 2
**Empirical Novelty And Significance:** 2
**Recommendation:** 5

**Clarity, Quality, Novelty And Reproducibility:**

The paper is well-written and organized well. I am primarily concerned with the novelty of the contribution, as it is very similar to existing algorithms as CQL or COMBO.

**Strength And Weaknesses:**

Strengths:

(1) The paper has a clear organization and is well-written.

(2) CSVE achieves impressive empirical performance against strong offline RL baselines.

Weaknesses:

(1) The algorithm itself is very similar to existing conservative offline RL algorithms, particularly CQL and COMBO. To my understanding, the authors do demonstrate theoretical and empirical advantages over CQL but not COMBO. In my opinion, COMBO is more similar as both CSVE and COMBO are model-based in the sense that they require learning a dynamics model.

(2) Though the backbone of the CSVE algorithm is clear, the authors add several extensions such as learning an ensemble of transition models (which COMBO does not do), and using a separate policy extraction step rather than alternating value- and policy-improvement as CQL, COMBO do. It is not clear from the empirical analysis whether those changes are important to the empirical performance, or the fact that the penalty is over states rather than state-action pairs (which is the central novelty of the proposed algorithm). The authors should consider such ablations, as well as a comparison to COMBO, in the experiments to make their results more compelling.

**Summary Of The Paper:**

The authors propose a new algorithm--CSVE--that learns conservative estimates of state value functions by penalizing values of OOD states. The authors prove that the estimated state value functions are lower-bounds of the true value in expectation over any state distribution. Finally, the authors evaluate their algorithm against state-of-the-art offline RL baselines.

**Summary Of The Review:**

To my understanding, the key idea of the proposed CSVE algorithm is to consider a penalty over states rather than state-action pairs. However, CSVE also consider some differences in algorithm design (see (2) in weaknesses) to similar baselines, so it is difficult to judge how important this key idea is to the improved performance. I believe the paper could be drastically improved by removing theses extensions as ablations, as well as considering COMBO as an additional baseline. I feel that CSVE and COMBO are much more closely related than CSVE and CQL, as the former are both model-based algorithms.

---

> ### Author Response · Authors · 2022-11-19
> **Response to Reviewer KLUN**
>
> Thank you for your constructive comments! We have written a comprehensive comparison of CSVE to CQL and COMBO, from new theoretical results, practical algorithm design and experimental performance improvement. We believe CSVE's novel contributions are obvious and not simply incremental to prior work.
>
> **Concern 1: Comparison with CQL and COMBO.**
> >The algorithm itself is very similar to existing conservative offline RL algorithms, particularly CQL and COMBO. To my understanding, the authors do demonstrate theoretical and empirical advantages over CQL but not COMBO. In my opinion, COMBO is more similar as both CSVE and COMBO are model-based in the sense that they require learning a dynamics model.
>
> We have discussed in details the difference and relation between CSVE and COMBO in the general responses. Here we briefly restate their differences as follows. First, both COMBO and CSVE guarantee the lower bound on expectation of state values, however CSVE is under more general state distribution. Second, CSVE directly penalize values of OOD states while COMBO penalizes values of OOD state-action pairs. Third, COMBO uses model to generate rollouts and interpolate with the existed dataset, while CSVE uses model to sample OOD states that are probably reachable by one-step from the data. In other words, COMBO extend the dataset by model rollouts, while CSVE does not. Fourth, in practical implementation, besides the similar choice of implementing dynamic model in the ensemble of deep neural networks, CSVE and COMBO are different on critic training and policy extraction.
>
> **Concern 2: Extensions in algorithm implementation**
> > "extensions such as learning an ensemble of transition models (which COMBO does not do)"
>
> We double checked and verified that COMBO also uses such an approach. Recently, deep ensemble (i.e., ensemble of deep neural networks with different initialization) has become the de facto distributional predictive model in model-based RL and beyond, for its good capability on capturing both aleatory and episdemic uncertainty.
>
>
> > "extensions such as .. using a separate policy extraction step rather than alternating value- and policy-improvement as CQL, COMBO do.""
>
> We indeed adopt the similar alternating value- and policy-improvement process too, where we use AWR with a exploration term in the policy-improvement process. It is described in Algorithm 1. Sorry that for the paper lenght limit, we put the algorithm in the appendix.
>
> > "The authors should consider .. a comparison to COMBO"
>
> We have compared with COMBO on D4RL mujoco v2, as shown in Table A1 in the common response. Besides, we are actively trying to rerun the COMBO official code, as Appendix D.4 of our revised paper. The challenge is that the official code behaves not stable, while so far there are no open-source COMBO implementation that can achieve the same or close level of performance as the paper.
>
> > performance effect of "a penalty over states rather than state-action pairs"
>
> We have added the new experiments to analyze this effect, by replacing the penalty on states with that on state action pairs. As shown in Appendix D.2, compared to penalizing Q, penalzing V leads to better and more stable aymptotic performance in experiemnts of D4RL mujoco v2 medium tasks.

---

### Official Review · Reviewer_uGHe · 2022-10-30

**Confidence:** 4
**Correctness:** 2
**Technical Novelty And Significance:** 1
**Empirical Novelty And Significance:** 2
**Recommendation:** 3

**Clarity, Quality, Novelty And Reproducibility:**

Overall, the paper is clear, but there are quite a few typos, etc, which can be fixed. Novelty: I don't think the paper is novel. I think that the method in the paper is of high quality, but it already exists in prior work if I am understanding properly.

**Strength And Weaknesses:**

Strength: The method is clearly intuitive and a good idea to do, so that's a plus.

Weaknesses:

- Many design decisions, no clear explanation: There are many design decisions in the paper that have not been ablated or explained in more detail -- for example, why AWR, and not just standard SAC/CQL-like policy extraction? why learn separate V(s) and Q(s, a), and not just directly penalize Q(s, a) on OOD states?

- Missing comparisons: the method uses a dynamics model to obtain new states, yet the method doesn't compare to COMBO? Seeing the resutls, seems like COMBO would perform better than this, which means that the results are not significant...

- Novelty: It seems like if the Q(s, a) were directly penalized on OOD states, and policy actions, the method in this paper would just be COMBO identically, and indeed COMBO is just a more generalization of this approach. So, unless the choice of V(s) can be justified rigorously, I don't think this paper provides a novel method.

**Summary Of The Paper:**

The paper proposes a CQL-like method for penalizing out-of-distribution states in offline RL. The method first describes an approach for off-policy evaluation with OOD state penalization, and then incorporates into the training pipeline for Q-functions, and then uses it for offline RL.

**Summary Of The Review:**

In all, I am not convinced if this paper is doing anything more than COMBO. Moreover, the paper doesn't compare to COMBO, and doesn't justify design decisions. In all, this makes me think that the paper is not yet significant for the community, and must address these issues. Thus, I opt to give a reject score.

---

> ### Author Response · Authors · 2022-11-19
> **Response to Reviewer uGHe**
>
> We thank the reviewer for the valuable comments and will improve the revision accordingly! Per your concern, we have discussed in details CSVE's contributions on new theoretical results and methodology with comparison to both CQL and COMBO, as well as the principle of why penalizing on states are better than that of state-action pairs.
>
>
> **Concern 1: Design decisions**
> We will explain our design choices as follows.
> > Question: Why AWR, and not just standard SAC/CQL-like policy extraction?
>
> As for the policy extraction part, we use AWAC-like technique (AWR) instead of sac-like method, because we need to ultilize the KL divergence constraint on the learned policy. However, a sac-like policy extraction method can not guarantee this. Note that although CSVE gets good conservative V-function estimation, policy extraction still requires the help of Q or Advantage function. CQL and COMBO can use sac-like policy extractor since they utilize a regularizer in their implementation, e.g., see Section 3.2 in CQL. However, we do not impose this constraint or regularizer. Thus we can not use standard SAC/CQL-like policy extraction.
>
> > Question: Why learn separate V(s) and Q(s,a) rather than directly penalize Q(s, a) on OOD states?
>
> On one hand, as suggested by the theory part (section 3), CSVE penalizes OOD state values directly rather than through Q(s, a). The details of our motivation is restated in the former common response. On the other hand, although CSVE intends to penalize OOD state values, we still require Q (or advantage) to recover the learnt policy; thus our implementation adopts separate Q and V.
>
> **Concern 2: Comparison with COMBO**
> We agree that it is necessary to compare with COMBO, and have *added a detailed section in the common responses*. CSVE satisfies that over the state distribution that samples penalizing states, it lower bounds the true values in expectation; while for COMBO, this distribution is constrained to be the initial distribution. Besides, our method can guarantee that over the marginal state distribution of data, it is no more than the true values in expectation plus a constant decided by sampling error, without the assumption of a big enough parameter($\alpha$); while COMBO lower bounds expectiation of the value over the initial state distribution with the assumption of constraint on $\alpha$.
>
> Though these two methods are all model-based, CSVE and COMBO leverage the model in a very different way -- CSVE uses the model to sampling OOD states while COMBO uses the model to augment the data in an interpolation way. Besides, as presented in Table A1 (in the common response), we found that generally CSVE performs better than COMBO in D4RL tasks.
>
> **Concern 3: Novelty over COMBO**
> We have added a separate common response to discuss CSVE's novelty over CQL and COMBO comprehensively.
>
> > "It seems like if the Q(s, a) were directly penalized on OOD states, and policy actions, the method in this paper would just be COMBO identically, and indeed COMBO is just a more generalization of this approach."
>
> It is true for COMBO which uses the model to do multi-step rollouts and penalize on them. However, it is not true for CSVE. Indeed, it is the direct penalization on OOD states make CSVE's lower bounds hold for any state distribution (i.e., any used policy's discounted state distribution), while COMBO's lower bounds hold on an initial state distribution. This makes CSVE decouple conservative state value estimation and policy, and thus we can design more flexible and efficient policy extractors. With respect to conservative value estimation, we think, CSVE is a generalization of COMBO. In the common response aprt, we have given more rigourous discussion on this.

---

### Decision · Program_Chairs · 2023-01-20

**Decision:**

Reject

**Justification For Why Not Higher Score:**

The method is very similar to COMBO, and contains many design decision that require more rigorous justification and ablation.  These require more experimental analyses, which will help to clarify the need of learning conservative state values and how it compares with IQL, CQL, and COMBO.

**Justification For Why Not Lower Score:**

N/A

**Metareview: Summary, Strengths And Weaknesses:**

This paper addresses the over-estimation problem offline reinforcement learning by proposing a conservative state value estimation.  Instead of penalizing reward or value estimation, CSVE directly penalizes out-of-distribution states, so that the estimated state value lower bounds the true values over the state distribution, and it is well upper bounded over the marginal state distribution.

The paper is well written, and addresses an important problem.  However, the method is very similar to COMBO, and contains many design decision that require more rigorous justification and ablation.  These require more experimental analyses, which will help to clarify the need of learning conservative state values and how it compares with IQL, CQL, and COMBO.

---

> ### Author Response · Authors · 2023-02-03
> **Post-review Comments**
>
> The decision is not a surprise, but it is sad that none of reviewers give any feedback after we submitted the comments and answered their concerns in details. We do hope that the AC could drive more discussion after the initial reviews. Best wishes.